# Comprehensive Simulation of Ca^2+^ Transients in the Continuum of Mouse Skeletal Muscle Fiber Types

**DOI:** 10.3390/ijms222212378

**Published:** 2021-11-17

**Authors:** Oscar A. Rincón, Andrés F. Milán, Juan C. Calderón, Marco A. Giraldo

**Affiliations:** 1Biophysics Group, Institute of Physics, University of Antioquia, Medellín 050010, Colombia; oandres.rincon@udea.edu.co; 2Physiology and Biochemistry Research Group-PHYSIS, Faculty of Medicine, University of Antioquia, Medellín 050010, Colombia; andres.milan@udea.edu.co (A.F.M.); jcalderonv00@yahoo.com (J.C.C.)

**Keywords:** ECC, Ca^2+^ dyes, mathematical simulation, muscle cells, tetanus

## Abstract

Mag-Fluo-4 has revealed differences in the kinetics of the Ca^2+^ transients of mammalian fiber types (I, IIA, IIX, and IIB). We simulated the changes in [Ca^2+^] through the sarcomere of these four fiber types, considering classical (troponin –Tn–, parvalbumin –Pv–, adenosine triphosphate –ATP–, sarcoplasmic reticulum Ca^2+^ pump –SERCA–, and dye) and new (mitochondria –MITO–, Na^+^/Ca^2+^ exchanger –NCX–, and store-operated calcium entry –SOCE–) Ca^2+^ binding sites, during single and tetanic stimulation. We found that during a single twitch, the sarcoplasmic peak [Ca^2+^] for fibers type IIB and IIX was around 16 µM, and for fibers type I and IIA reached 10–13 µM. The release rate in fibers type I, IIA, IIX, and IIB was 64.8, 153.6, 238.8, and 244.5 µM ms^−1^, respectively. Both the pattern of change and the peak concentrations of the Ca^2+^-bound species in the sarcoplasm (Tn, PV, ATP, and dye), the sarcolemma (NCX, SOCE), and the SR (SERCA) showed the order IIB ≥ IIX > IIA > I. The capacity of the NCX was 2.5, 1.3, 0.9, and 0.8% of the capacity of SERCA, for fibers type I, IIA, IIX, and IIB, respectively. MITO peak [Ca^2+^] ranged from 0.93 to 0.23 µM, in fibers type I and IIB, respectively, while intermediate values were obtained in fibers IIA and IIX. The latter numbers doubled during tetanic stimulation. In conclusion, we presented a comprehensive mathematical model of the excitation–contraction coupling that integrated most classical and novel Ca^2+^ handling mechanisms, overcoming the limitations of the fast- vs. slow-fibers dichotomy and the use of slow dyes.

## 1. Introduction

In mammalian skeletal muscle fibers, the action potentials (AP) lead to contractions mediated by the release of Ca^2+^ from the sarcoplasmic reticulum (SR), in a process known as excitation–contraction coupling (ECC) [1]. After release, the Ca^2+^ ions bind to a diversity of sites, which include troponin (Tn), parvalbumin (PV), and adenosine triphosphate (ATP). They also flow into the mitochondria (MITO) [1], before being transported back to the SR by a Ca^2+^ ATPase (SERCA). Small amounts of Ca^2+^ are transported outside the cell by the Na^+^/Ca^2+^ exchanger (NCX). Store-operated Ca^2+^ entry (SOCE) allows Ca^2+^ enter the fiber through Orai1, as a response to the intra SR Ca^2+^ sensing function of the stromal interaction molecules (STIM) [2,3,4]. In fast twitch fibers, SOCE acts in a transient fast mode during an individual AP and after each AP in a train of stimulations [5].

Experiments in muscle fibers loaded with fluorescent indicators have revealed sizeable differences in the kinetic parameters of electrically elicited Ca^2+^ transients according to the continuum of fiber types [6]. These variances arise due to the differential quantity and kinetics of the proteins and organelles involved in Ca^2+^ handling present in the muscle fibers [1,6]. For instance, SERCA is more abundant in fast fibers [7,8], and PV is almost negligible in slow, but abundantly found in the fastest fibers [9].

Mathematical models that integrate information obtained on mammalian ECC under different experimental conditions have been used to simulate Ca^2+^ transients in both slow-twitch and fast-twitch fibers [10,11]. However, the nature of the fibers used has not always been doubtlessly established. Furthermore, a major limitation of several models is their dichotomic approach (slow and fast), yet there are experimentally measured Ca^2+^ transients of at least four fiber types: I, IIA, IIX, and IIB [6]. Some models have not included important mechanisms dealing with Ca^2+^, such as the mitochondria or the NCX, despite their importance in shaping the Ca^2+^ transients in different fiber types [12,13].

A recent model of skeletal muscle ECC, interestingly included the MITO and proteins such as the mitochondrial Ca^2+^ uniporter (MCU) and the mitochondrial NCX (NCE) [14]. However, that model was performed based on Ca^2+^ transients’ measurements with Fura-2, while the most suitable dyes to study ECC in skeletal muscle seem to be the low affinity, fast Ca^2+^ dyes, such as Mag-Fura-2 and Mag-Fluo-4 [1,15]. This explains why the reported amplitude of the signal (0.4 µM) was very low compared to previously published values being 10–20 µM in murine Flexor digitorum brevis (FDB) fibers with Mag-Fura-2 and Mag-Fluo-4 [15,16,17].

In this work, we used measurements carried out with the fast Ca^2+^ dye Mag-Fluo-4 and a theoretical model to estimate the Ca^2+^ movements produced during single and tetanic Ca^2+^ transients in the four murine skeletal muscle fiber types. This permitted us to make a simulated comparison of the Ca^2+^ movement across the four fiber types, including the variations in the mitochondrial [Ca^2+^], the NCX, the SOCE, and their influence on the sarcoplasmic Ca^2+^ regulation, and further overcoming the limitations imposed by slow Ca^2+^ dyes.

## 2. Results

Experimentally recorded, raw, single, and tetanic Mag-Fluo-4 Ca^2+^ transients were calibrated in order to obtain the ∆[Ca^2+^] in the sarcoplasm (Figure 1). The [Ca^2+^] peaks for the continuum of fiber types were: IIB and IIX: 16.58 µM, IIA: 12.77 µM, and I: 10.13 µM (Figure 1A). For the tetanic Ca^2+^ transients, subsequent peaks were also calculated (I: 11.24, 12.95, 14.81, and 16.47 µM; IIB: 12.13, 12.29, 12.24, and 11.96 µM) (Figure 1B).

These calibrated signals fed all next estimations and simulations, as described in the Methods section and Table 1. First, we mathematically estimated the release rate of Ca^2+^ (*J*_Rel_) for both single and tetanic transients (Figure 2A,B; Table 2) and then simulated the Ca^2+^ kinetics in the sarcoplasm (Figure 2C,D), the SR (Figure 2E,F), and MITO (Figure 2G,H). Fibers type II peaked higher than fibers type I (~137% higher for IIA and 269–277% higher for IIX and IIB). The *J*_Rel_ estimated in tetanic Ca^2+^ transients shows that the last peak’s amplitude is reduced over 10 times for type I fibers and up to 7 times for IIB, IIX, and IIA (Figure 2B; Table 2). The simulated sarcoplasmic ∆[Ca^2+^] closely resembled the experimental recording described above. The [Ca^2+^]_SR_ rapidly decreased and slowly recovered as expected. Although qualitatively similar for all fiber types, quantitative differences arose mainly between the fibers type I and all fibers type II, in agreement with the fact that the Ca^2+^ released by the fibers type I was the lowest.

There was a higher value in ∆[Ca^2+^]_MITO_ for the fibers type I (up to four times) as compared to the three fibers type II. The total [Ca^2+^] in MITO, considering both forms, bound to B and free, achieves a peak value of 0.26–1.12 µM after a single AP and 0.72–2.8 µM after 5 AP. Regarding the SR compartment, our calculations show that the available Ca^2+^ is reduced up to ~63% (I: 82%, IIA: 63%, IIX: 64%, and IIB: 63%) for single transients, but to almost 53% (I: 76%, IIA: 63%, IIX: 56%, and IIB: 53%) after a train of five shocks (Figure 2E,F).

We also calculated the variations in [Ca^2+^] for the buffers present in the three simulated compartments (Figure 3), for both single (left column) and tetanic transients (right column). The differences in [CaPV] show the PV buffer influence in the first part of [Ca^2+^] decay obtained in the IIX and IIB fibers, compared with I and IIA fibers. The second part of the decay, when the sarcoplasmic [Ca^2+^] is near steady state, was less affected by the PV buffering. The Δ[CaDye] and Δ[CaATP] simulations were not shown as they resemble the shape of the sarcoplasmic [Ca^2+^] already shown (Figure 2C,D), but with different peaks (reported in Table 3). The ∆[CaDye] explains only 2.7% of the intracellular dye in the fiber type IIA and 2.2% in fiber types I, IIX, and IIB, thus ensuring dye unsaturation.

Figure 4 shows that the [Ca^2+^] in the sarcoplasm is mainly recaptured by the SERCA. In general, the amount of Ca^2+^ handled by the NCX was 41–121 times lower after 1 AP and 34–88 times lower after 5 AP, compared to the SERCA capacity. Moreover, the rate of transport by NCX was notably reduced when the sarcoplasmic [Ca^2+^] achieves low values in IIX and IIB fibers given its low affinity. The SERCA pump maintains its influence in [Ca^2+^] regulation near resting conditions. The [Ca^2+^] returned to the sarcoplasm by the SOCE was negligible in all fibers compared to the total [Ca^2+^] released. The total [Ca^2+^] handled by each mechanism after the simulated time intervals, the free [Ca^2+^] reached in each compartment, and the Ca^2+^ bound to the chemical species are reported in Table 3. A longer time interval of the tetanic Ca^2+^ transient simulation was included for some reactions (Appendix A). The total amount of Ca^2+^ obtained at rest considering all compartments of the model, including the extracellular space, remained constant during the simulated activation interval. We obtained that the variations in the total amount of Ca^2+^ were lower than 10^−6^ µM. This result evidence that truncation errors were negligible during the simulations.

## 3. Discussion

The main findings of the present work were: (i) during a single twitch, the sarcoplasmic peak [Ca^2+^] for fibers type IIB and IIX is between 15–25 µM, and for fibers type I and IIA reaches 6–12 µM, (ii) both the pattern of change and the peak concentrations of the Ca^2+^-bound species in the sarcomere, the sarcolemma, and inside the SR showed the order IIB ≥ IIX > IIA > I, (iii) the mitochondrial peak [Ca^2+^] and the MITO buffers saturation showed the pattern I >> IIA >> IIX ≥ IIB.

Previous models of mammalian ECC were affected by either uncertainty in the classification of fiber types, unreliable kinetics of the raw Ca^2+^ signals due to the use of slow Ca^2+^ dyes, or the lack of information about the role of several intracellular compartments in Ca^2+^ handling. To overcome these limitations, we based our model on the first calibration of Ca^2+^ transients of the four main fiber types found in mammals, obtained using the fast Ca^2+^ dye Mag-Fluo-4, and integrated new information gathered on MITO, SR, NCX, and SOCE, along basic knowledge on sarcoplasmic Ca^2+^ movements and buffering.

### 3.1. A Model which Includes Four Fiber Types

Preceding models addressed ECC in one or two fiber types, mainly as most previous functional and biochemical information came from a dichotomic approach of muscle fibers: either slow vs. fast, or type I vs. type II [13,14,16,34,35,36,37]. Additionally, simulations that have recently become spatially and mathematically more complex have remained biochemically oversimplified [38,39]. By taking Ca^2+^ transients obtained in molecularly typed fibers covering the whole spectra from I to IIB as experimental source, our model adds novel information on the ECC-fiber type relationships.

The fact that molecular and biochemical differences (i.e., isoforms and their biochemical properties) underlie the differences in ECC among fiber types has been described in detail elsewhere and we refer the readers to those papers [1,6,40].

However, there is still a lack of molecular and biochemical information, particularly for fibers type IIA and IIX. Our adjustments and results for these two types of fibers seem reliable as most values obtained laid between those of fibers type I and IIB. This agrees with the fact that fibers IIA and IIX showed kinetics of the Ca^2+^ transients, Ca^2+^ sensitivity, and other dynamic properties which are mostly intermediate between I and IIB [6,40,41,42]. Together, this confirms that most values of molecular, biochemical, and physiological parameters follow a continuum from I, to IIA, to IIX, to IIB.

### 3.2. [Ca^2+^] Kinetics with a Fast Ca^2+^ Dye

The Ca^2+^ transients modeled in the present study were obtained with the fast Ca^2+^ dye Mag-Fluo-4. Two issues with quantitative impact on the results deserve attention: the biochemical properties of the dye and its loading conditions.

This dye has a 2:1 Mag-Fluo-4–Ca^2+^ binding stoichiometry, with an in situ *K*_d_ of 1.652 × 10^5^ µM^2^ [17]. This explains why it can reliably track the Ca^2+^ transients even in the fastest types, i.e., IIB, demonstrate subtle differences among all four fiber types and resolve every peak in a tetanic transient, being a trustable source for the model. On the other hand, as the fibers typically have ~200 µM of Mag-Fluo-4 [17], we found that less than ~3% of the dye is bound to Ca^2+^.

The very low affinity of the dye and the lack of saturation confer Mag-Fluo-4 the ability to determine a trustable peak sarcoplasmic [Ca^2+^]. Our results with Mag-Fluo-4 (calibration performed in the present study and [17]) join those with Mag-Fura-2 [16,35] which show that the peak [Ca^2+^] in fast fibers IIX or IIB is typically between 15 and 25 µM in mammalian skeletal muscle (16–22 °C). The numbers for type I and IIA fibers also agree with a study reporting peak Ca^2+^ for soleus slow fibers [13]. That work, however, may have really used type I and IIA fibers, as the periphery of the soleus muscles of mouse used in the experiments have both types of fibers evenly distributed, as we have verified using specific antibodies and confocal microscopy (not shown). In conclusion, that paper and our results agree in that peak [Ca^2+^] for fibers types I and IIA is between 6 and 12 µM [13]. The above numbers are one order of magnitude higher than those reported with slow dyes [14,33,43,44]. This fact may, for instance, reduce the estimated maximum rate of Ca^2+^ flux during SOCE or excitation-coupled Ca^2+^ entry (ECCE) activations [33,45], as this value depends on the driving force for Ca^2+^ [46], which in turn is reduced if sarcoplasmic [Ca^2+^] is raised. Trustable peak sarcoplasmic [Ca^2+^] also permits a better quantitation of the chemical species involved in ECC ([CaTn], [CaTNS], [CaPV], [CaATP], [CaDye], free [Ca^2+^], etc.), including peak [Ca^2+^] inside compartments such as MITO.

### 3.3. Comprehensive Integration of Mechanisms Involved in Ca^2+^ Handling: Sarcoplasm, SR, MITO, NCX and SOCE

The ECC is becoming more complex [1] and experimentally addressing some questions is challenging. For instance, important papers have investigated Ca^2+^ kinetics in MITO and SR, as well as SOCE function in fast fibers [33,47,48], but differences among all fiber types are infrequently [12] or never studied. The mathematical model presented here helps address these limitations.

The peak flux differences found in the release rate are consistent with the two- to ten-fold higher content of the ryanodine receptor (RyR) in the fastest fibers, compared to the other fibers [6,49,50,51]. The comparable values of half-width and rise time in all fiber types can be explained as all fiber types share the RyR1 isoform [52].

Upon release, Ca^2+^ has multiple fates: (i) binds to classical sarcoplasmic buffers such as PV, Tn, TNS, ATP, and the dye; (ii) enters into MITO and SR, or (iii) is recycled through the NCX and SOCE mechanisms [1].

In fast fibers (16–22 °C), [CaTn] has ranged from 80 µM [27] to around 240 µM [16,39], being ~199 µM for IIA, IIX, and IIB in the present study, within 7 ms from the beginning of the release of Ca^2+^. Slow fibers gave values of 85 µM [13] and ~110 µM (fibers type I in the present study) within 15 ms. Different reaction rate constants arising from temperature corrections, our larger SERCA capacity, as well as the inclusion of MITO and SOCE, explain why the peak [CaTn] is somewhat lower in the fast fibers in our model than in the model of Baylor and Hollingworth (2007) [16]. The TNS peaked 31 µM during a train of APs in slow fiber types after 200 ms [13], and in the present work, such as with Tn, have peaked below, being 22 µM. Specifically in type IIX and IIB fibers, which have a higher quantity of PV, their influence is reduced, as the binding and unbinding rates are similar to those of PV.

The peak concentration of [CaPV] in fast fibers has ranged from 90 µM [27] to 120 µM [16], close to the 116–142 µM for IIX and IIB found in the present study, after 25 ms of Ca^2+^ release. Slow fibers are devoid of PV [13], or very low values have been measured in fibers type I and IIA in murines [53,54], justifying the values found (~0.5-5 µM) here. Then, the continuum of [CaPV] in I, IIA, IIX, and IIB fibers followed the order IIB > IIX >> IIA > I.

The peak [CaATP] we found for fibers IIX and IIB coincides with the values reported previously for fast fibers, i.e., 60 µM [27] and ~70 µM [16]. In all cases, the peak was within 6 ms from the release. In type I fibers, we report a value of 23 µM, with IIA being an intermediate. Then, the continuum of [CaATP] in all fibers was IIB = IIX > IIA > I.

The [CaDye] concentration reached a peak at around 22 µM [16] or 5 µM (IIB, present study) for fast fibers, indicating far-from-saturation conditions and likely not affecting Ca^2+^ release. Variations depend on differences in loading conditions used by the researchers in different laboratories. The value found for type I fibers in the present study was 3 µM. The order resembled that of the peak [Ca^2+^] in the sarcoplasm and the Ca^2+^-ATP reaction.

Peak change in [Ca^2+^] inside MITO was reported to be 0.35 µM for a single twitch in fast fibers in Marcucci et al. (2018) [14]. The peak [Ca^2+^] was ~0.25 µM in our results for fibers type IIX and IIB, about 25 ms after Ca^2+^ release. This concentration is similar to that obtained experimentally in fast mouse fibers [47]. Somewhat later, [Ca^2+^] peaked inside MITO in type I fibers, reaching a value of 0.9 µM. This delay likely reflects Ca^2+^ diffusion into MITO, as measured in mouse using genetically encoded sensors [55]. Notably, the peak [Ca^2+^] was far higher in I than in II, likely as the MCU of slow fibers has more capacity than that of the fast fibers [30], with no difference in the properties of the NCE, allowing more Ca^2+^ to accumulate inside MITO of fibers type I. Additionally, in fibers type I, the lower amount of PV compared to II allows more Ca^2+^ free to diffuse into MITO. For fibers IIA, their intermediate amount of Ca^2+^ released, along with their intermediate amount of PV and SERCA content may explain their midsize value of Ca^2+^ inside MITO. According to their peak amplitude, Ca^2+^ transients in MITO can be ordered as I >> IIA >> IIX ≥ IIB.

Previous works have estimated the total concentration of MITO buffers [B]. From the simulated analysis of the total [B] in cardiac muscle fibers performed in [56], a value of 2 µM was found to be optimal. In the work of Marcucci et al. (2018) [14] for skeletal muscle fibers a larger buffer concentration of 20 µM was also tested. In the present study, a value of 20 µM was used. A sizeable total [B] inside MITO is expected in order to deal with the larger and faster amounts of Ca^2+^ released in skeletal compared to cardiac muscle, and to explain the slowing of decay of the Ca^2+^ transients when the MITO uptake is blunted [12,47]. However, well calibrated, high resolution, experimental measurements of MITO Ca^2+^ buffering and kinetics are still required to reach an accurate estimate of the total [B] and [Ca^2+^] inside MITO in skeletal muscle fibers.

Ca^2+^ pumped by the SERCA was estimated to be 1.5–3.5 µM in fast fibers [16], but amounted up to ~50 µM for IIX fibers in our model, after 25 ms. In slow fibers, a value somewhat below 1 µM was reported [13] at the same time (50 ms), but we found ~15 µM for type I. The differences between models can be explained due to temperature, as well as half-width differences in the recordings that fed the simulation. According to the intermediate kinetics of the decay phase of the Ca^2+^ transients in fibers IIA, as well as our previous discussion, midsize values for IIA were expected. The difference of Ca^2+^ pumped by SERCA 50 ms after Ca^2+^ release was IIB = IIX >> IIA >> I. The total capacity of Ca^2+^ extrusion by the NCX in fibers type IIB, IIX, and IIA was only 1.1–-1.3 times higher than in I, suggesting that the larger amount of NCX1 in fibers type I is balanced by the larger capacity of the NCX3 present in fibers type II. The Ca^2+^ extruded by the t-system seems to be immediately recycled and has been called a counter-flux [5]. Our results confirmed that SOCE is quantitatively small in skeletal muscle [33,46], and it is difficult to speculate on the importance of the minor differences among fibers. More robust experimental data should be gathered before a conclusion about this issue can be stated.

### 3.4. Final Remarks

Although the above analyses give averages of peak sarcoplasmic/compartments [Ca^2+^], spatially refined models for fast fibers have shown up to a 20-fold gradient in the sarcoplasmic [Ca^2+^], depending on the distance of a subcellular region from the Ca^2+^ release units [13,16,39]. This phenomenon is also expected to apply to all fiber types, but the magnitude of those gradients inside the fibers was not explored in the present study.

Single-compartment simulations are expected to have errors associated with an inability to estimate local gradients in [Ca^2+^] and in the [Ca^2+^] bound to the binding sites. In the present study, the SOCE flux is associated with changes in [Ca^2+^] in the SR compartment. However, Ca^2+^ gradients in the SR have been measured during ECC and the Ca^2+^ levels decrease more rapidly in the terminal cisternae than in other regions of the SR [57]. Therefore, a model that considers the gradients in the SR would allow a more accurate estimate of the amount of Ca^2+^ entered by the SOCE.

Additionally, as our Ca^2+^-bound chemical species had higher concentrations than those recently estimated, the thermal changes associated with ECC in mammalian muscle should be higher than proposed [27]. Our model may be a source to build a more complete model on thermal changes in all fiber types during single and tetanic stimulation.

## 4. Materials and Methods

### 4.1. Experimental Single and Tetanic Ca^2+^ Transients

Typical fluorescence experimental recordings (*F*) of single Ca^2+^ transients of fiber types I, IIA, IIX, and IIB were taken from [6]. Tetanic Ca^2+^ transients (100 Hz) of types I and IIB fibers were taken from [12]. The tetanic Ca^2+^ transient of IIA and IIX fibers were simulated assuming that they share morphology with I and IIB fibers, respectively, as published [6,12]. In all cases the signals were obtained with Mag-Fluo-4 in electrically stimulated isolated fibers from mouse muscles. Fibers were classified based on myosin heavy chain determination as detailed described in the above references. The conversion of *F* of single and tetanic experimental recordings to [Ca^2+^] was performed according to the calibration method and the value parameters presented in [17]. Briefly, [Ca^2+^] was calculated using the expression:(1)$$\left[{\mathrm{Ca}}^{2+}\right]=\frac{K_{d}}{{\left[D\right]}_{T}}\frac{\left({F-F}_{\min}\right)\left(F_{\max}-F_{\min}\right)}{2{\left(F_{\max}-F\right)}^{2}}$$
where *F_max_* and *F_min_* are the maximum and minimum fluorescences (150.9 A.U and 0.14 A.U) respectively, *K_d_* is the in situ dissociation constant of Mag-Fluo-4 (1.652 × 10^5^ µM^2^), and [D]_T_ is the total dye concentration (229.1 µM), at 20 °C. As the experimental Ca^2+^ transients that feed the model were acquired between 21–23 °C, no temperature corrections were performed in the calibration parameters.

### 4.2. Model Description

The rate of change of free [Ca^2+^] was described in three compartments (SR, sarcoplasm, and MITO) with a system of differential equations. The rate of change of [Ca^2+^] with respect to time in the SR was determined by
(2)$$\frac{d{\left[{\mathrm{Ca}}^{2+}\right]}_{\mathrm{SR}}}{\mathrm{dt}}{=-J}_{\mathrm{Rel}}\;{+J}_{\mathrm{SERCA}}-F\left({\left[{\mathrm{Ca}}^{2+}\right]}_{\mathrm{SR}},\;\mathrm{CSQ}\right)$$
where *J*_Rel_ is the release rate flux, *J*_SERCA_ is the SERCA flux, and *F* the reaction rate of Ca^2+^ in the SR with calsequestrin (CSQ). The rate of change of sarcoplasmic [Ca^2+^] was expressed as:(3)$$\frac{d\left[{\mathrm{Ca}}^{2+}\right]}{\mathrm{dt}}{=J}_{\mathrm{Rel}}-J_{\mathrm{SERCA}}-J_{\mathrm{NCX}}-\frac{d{\left[{\mathrm{Ca}}^{2+}\right]}_{\mathrm{MITO}}}{\mathrm{dt}}{+J}_{\mathrm{SOCE}}-F\left({\left[{\mathrm{Ca}}^{2+}\right]}_{\mathrm{SR}},\;S\right)$$
where *J*_NCX_ is the NCX flux, [Ca^2+^]_MITO_ the mitochondrial [Ca^2+^], *J*_SOCE_ is the SOCE flux, and *F*([Ca^2+^], S) the reaction rate with the sarcoplasmic Ca^2+^ binding sites S. The change in time of [Ca^2+^] in the MITO was given by
(4)$$\frac{d{\left[{\mathrm{Ca}}^{2+}\right]}_{\mathrm{MITO}}}{\mathrm{dt}}{=J}_{\mathrm{MCU}}-J_{\mathrm{NCE}}-F\left({\left[{\mathrm{Ca}}^{2+}\right]}_{\mathrm{MITO}},\;B\right)$$
where *J*_MCU_ was the inflow through the MCU, *J*_NCE_ outflow through the NCE, and *F*([Ca^2+^]_MITO_, B) the reaction rate with B. The B was 20 µM of binding sites, which react with Ca^2+^ [14].

The parameters that describe the *J_Rel_*, and the values of maximum capacity of SERCA, NCX, NCE, and SOCE (*V*_SERCA_, *V*_NCX_, *V*_NCE_, and *V*_SOCE_), were fitted to the measured Ca^2+^ transients. All other values of the model were taken from the literature, and the specific references are given in Table 1 and Appendix A.

#### 4.2.1. Release Rate of Ca^2+^

The *J_Rel_* of Ca^2+^ from the SR in fast and slow fibers were estimated from measurements of sarcoplasmic Ca^2+^ transients obtained with Mag-Fluo-4 (Figure 1A), as previously done with Mag-Fura-2 [35]. This approximation assumes that Ca^2+^ and other chemical species that react with Ca^2+^ are uniformly distributed in the sarcoplasm, which is thus described as a single compartment. The time derivative of the total Ca^2+^ concentration in the sarcoplasm, *d*[Ca]_T_/*dt*, is given by the sum of the inflows and outflows through the membrane. Therefore, from the sum of *d*[Ca]_T_/*dt* and outflows, we obtained the release rate of Ca^2+^. A gaussian model was used to simulate the *J_Rel_* elicited by a single AP:(5)$$J_{\mathrm{Rel}}\left(t\right)=\overset{M}{\underset{j=1}{\sum }}f_{\mathrm{Rel},j}\overset{N}{\underset{i=1}{\sum }}R_{i}e^{-{\left(\frac{{(t-(j-1)T}_{2})-T_{1,i}}{\tau_{i}}\right)}^{2}}$$
where *N* is the number of peaks used to fit the *J_Rel_* produced by a single AP, *R* is the peak amplitude, *T**_i_* is the location in the time axis, and *τ* relates to the width of each peak. *R*, *T*, and *τ* were adjusted to fit the peak amplitude, half-width, rise time, and decay time of the release rate of Ca^2+^ elicited by a single AP (Table 2). *f_Rel_* is the multiplication factor used in the second and subsequent APs to fit the measured peaks amplitudes. *M* is the number of APs and *T*_2_ of 10 ms is the time between stimulations.

#### 4.2.2. Reaction of Ca^2+^ with Sarcomeric Buffers

The local reactions of Ca^2+^ with S of the chemical species in the compartments were described by the law of mass action,
(6)$$F\left({[\mathrm{Ca}}^{2+}],\;S\right)=k_{-}\left[{\mathrm{SCa}}^{2+}\right]-k_{+}\left[{\mathrm{Ca}}^{2+}\right]\left[S\right]$$
with *k*_+_ the binding rate constant, and *k_−_* the unbinding rate constant. The interactions of Ca^2+^ and Mg^2+^ with the binding sites were described as reversible reactions. The reactions with ATP, B and CSQ were described as single reversible reactions. As Mg^2+^ competes with Ca^2+^ for the PV, their interactions are described with two simultaneous reversible reactions. Additionally, as recently described in Milán et al. (2021) [17], two molecules of the fluorescent indicator Mag-Fluo-4 (Dye) bind to one Ca^2+^. The *k*_+_ and *k_−_* values associated with the ATP, Tn, and PV binding sites were taken from Baylor and Hollingworth (2007) [16] and Hollingworth et al. (2012) [13], CSQ from Barclay et al. (2021) [27], and B from Marcucci et al. (2018) [14]. A temperature of 22 °C and a Q_10_ of 2 were considered to adjust the reaction rates (Appendix A). The resulting ordinary differential equations system was solved using the *ode15s* solver in MATLAB 2021b (MathWorks, Natick, MA, USA). The occupancy fraction of the binding sited at equilibrium were obtained from the simulated data.

#### 4.2.3. Muscle Proteins Concentration

The Tn molecules concentration were 120 µM in all fiber types such as in Hollingworth et al. (2012) [13]. As each fast fiber Tn molecule has two Ca^2+^ binding sites and slow fiber molecules one, we thus assumed a concentration of 240 µM with positive cooperativity for IIA, IIX, and IIB fibers, and 120 µM for type I fibers. Additionally, as each Tn molecule has two non-specific sites (TNS) we considered concentrations of 240 µM for the binding sites of fast and slow fibers. As the Tn molecules are located in the myofibrillar space (MS), the average [Tn] and [TNS] in the sarcoplasm was rescaled by the occupation of the MS volume in the sarcoplasm (*V*_MS_ and *V*_sarc_ in Table 1).

The [PV] in the fastest fibers can achieve large concentrations, 1.5–1.9 mM [16,27]. We also considered that fibers type I have 300 times less [PV] [7], leading to 6 µM. Fibers IIA have a [PV] closer to fibers type I than type IIB, so we assumed 10 times more PV content in IIA than in I type [58], giving 60 µM, which is far lower than IIX and IIB, as expected. The ATP, Dye, CSQ, and B concentrations were listed in Table 1.

#### 4.2.4. Reuptake Rate of Ca^2+^ by SERCA

The reuptake of Ca^2+^ by SERCA to the SR is described by the Michaelis–Menten kinetics and expressed as:(7)$$J_{\mathrm{SERCA}}=V_{\mathrm{SERCA}}\frac{{\left[{\mathrm{Ca}}^{2+}\right]}^{h_{\mathrm{SERCA}}}}{{\left[{\mathrm{Ca}}^{2+}\right]}^{h_{\mathrm{SERCA}}}+K_{\mathrm{SERCA}}^{h_{\mathrm{SERCA}}}}$$
where *V*_SERCA_ is the maximum flux rate, [Ca^2+^] is the Ca^2+^ concentration in the sarcoplasm, *K*_SERCA_ is the dissociation constant and *h*_SERCA_ the Hill coefficient. The SERCA isoform predominantly expressed in fast-twitch muscle fibers is the 1a, whereas in slow-twitch is the 2a [59]. Both have similar *K*_SERCA_ and *h*_SERCA_ [29], however different content of SR pump molecules [8], and thus *V*_SERCA_. The *V*_SERCA_ was then adjusted to match the decay phase of the measured and simulated Ca^2+^ transients in all fiber types.

#### 4.2.5. The Mitochondrial Ca^2+^ Uniporter (MCU) Inflow and the Sodium Ca^2+^ Exchanger (NCE) Outflow

The *J*_MCU_ was represented by a saturable first-order transporter, independent of the internal [Ca^2+^]_MITO_:(8)$$J_{\mathrm{MCU}}=V_{\mathrm{MCU}}\frac{{\left[{\mathrm{Ca}}^{2+}\right]}^{h_{\mathrm{MCU}}}}{{\left[{\mathrm{Ca}}^{2+}\right]}^{h_{\mathrm{MCU}}}+K_{\mathrm{MCU}}^{h_{\mathrm{MCU}}}}$$
where *V*_MCU_ is the maximum flux rate, [Ca^2+^] is the Ca^2+^ concentration in the sarcoplasm, and *K*_MCU_ is the [Ca^2+^] where the transport rate is half-maximum. *V*_MCU_ values were taken as 0.012 μM ms^−^^1^ in fast and 0.049 μM ms^−^^1^ in slow fibers at 16 °C [13,16], and adjusted considering a Q_10_ of 2.

The *J*_NCE_ was modelled with a stoichiometry of 3:1 and described with the following expression [31]:(9)$$J_{\mathrm{NCE}}=V_{\mathrm{NCE}}(\frac{e^{\frac{0.5∆\Psi_{m,\mathrm{mito}}F}{\mathrm{RT}}}\frac{{\left[{\mathrm{Na}}^{+}\right]}^{3}{\left[{\mathrm{Ca}}^{2+}\right]}_{\mathrm{mito}}}{K_{{\mathrm{NCE},\mathrm{Na}}^{+}}^{3}\;\cdot \;K_{{\mathrm{NCE},\mathrm{Ca}}^{2+}}}-e^{-\frac{0.5∆\Psi_{m,\mathrm{mito}}F}{\mathrm{RT}}}\frac{{\left[{\mathrm{Na}}^{+}\right]}_{\mathrm{mito}}^{3}\left[{\mathrm{Ca}}^{2+}\right]}{K_{{\mathrm{NCE},\mathrm{Na}}^{+}}^{3}\;\cdot \;K_{{\mathrm{NCE},\mathrm{Ca}}^{2+}}}}{1+\frac{{\left[{\mathrm{Na}}^{+}\right]}^{3}}{K_{{\mathrm{NCE},\mathrm{Na}}^{+}}^{3}}+\frac{{\left[{\mathrm{Ca}}^{2+}\right]}_{\mathrm{mito}}}{K_{{\mathrm{NCE},\mathrm{Ca}}^{2+}}}+\frac{{\left[{\mathrm{Na}}^{+}\right]}^{3}{\left[{\mathrm{Ca}}^{2+}\right]}_{\mathrm{mito}}}{K_{{\mathrm{NCE},\mathrm{Na}}^{+}}^{3}K_{{\mathrm{NCE},\mathrm{Ca}}^{2+}}}+\frac{{\left[{\mathrm{Na}}^{+}\right]}_{\mathrm{mito}}^{3}}{K_{{\mathrm{NCE},\mathrm{Na}}^{+}}^{3}}+\frac{\left[{\mathrm{Ca}}^{2+}\right]}{K_{{\mathrm{NCE},\mathrm{Ca}}^{2+}}}+\frac{{\left[{\mathrm{Na}}^{+}\right]}_{\mathrm{mito}}^{3}\left[{\mathrm{Ca}}^{2+}\right]}{K_{{\mathrm{NCE},\mathrm{Na}}^{+}}^{3}K_{{\mathrm{NCE},\mathrm{Ca}}^{2+}}}})$$
where ∆Ψ_m,mito_ is the mitochondrial membrane potential, *V*_NCE_ is the NCE activity, and *K*_NCE,Ca_^2+^ and *K*_NCE,Na_^+^ are the dissociation constants for the Ca^2+^ and Na^2+^ binding to the NCE, respectively. *F*, *R*, and *T* denote the Faraday constant, the ideal gas constant and temperature, respectively. *V*_NCE_ was modulated in the fastest fibers to reproduce the speed of the [Ca^2+^]_mito_ decay phase, completed in a period of about 100 ms and measured during a single twitch, as in Rudolf et al. (2004) [55].

#### 4.2.6. Ca^2+^ Flux through the NCX

To describe the sarcolemmal NCX, the same expression of the NCE, with stoichiometry of 3:1 was used, although with different simulated values. Reported Ca^2+^-transport rate values for SERCA range from 10 (in membrane vesicles) to 13–14 nmol mg⁻^1^ min⁻^1^ (NCX3 and NCX1 isoforms, respectively) [60,61]. We assumed a maximum transfer rate (*V*_NCX_) between 30 and 40% higher than that of SERCA. As the NCX is predominantly located in the T-tubule membrane [62], whereas the SERCA in the SR membrane, we thus rescaled *V*_NCX_ by a ratio (T-tubule membrane/SR membrane) of 0.066 and 0.111 in fast and slow fibers [20], i.e., between 214.5 and 120.1 µM s^−1^ for all fibers type II. In fibers type I we used 93.2 µM ms^−1^. For *K*_NCX,Ca_^2+^ and *K*_NCX,Na_^+^, we, respectively took values of 140 µM and 14 mM in fast fibers, and 130 µM and 11 mM in slow fibers [32].

#### 4.2.7. The Effect of Store-operated Ca^2+^ Entry (SOCE)

We described the flux, *J*_SOCE_, as a fraction, *P*_SOCE_, of a given maximum value, and *V*_SOCE_ as *J*_SOCE_= *V*_SOCE_
*P*_SOCE_ [63]. As said, *P*_SOCE_ change as a function of the [Ca^2+^] in the SR, which can be described by the function:(10)$$P_{\mathrm{SOCE}}=\frac{K_{\mathrm{SOCE}}^{h_{\mathrm{SOCE}}}}{{\left[{\mathrm{Ca}}^{2+}\right]}_{\mathrm{SR}}^{h_{\mathrm{SOCE}}}{+K}_{\mathrm{SOCE}}^{h_{\mathrm{SOCE}}}}$$
where *K*_SOCE_ and *h*_SOCE_ are the Ca^2+^ dissociation constant of STIM1 (0.35 mM) and the Hill coefficient (4.7), respectively [33]. We assumed that the *V*_SOCE_ in the fastest IIB fibers achieved up to 35 µM s^−^^1^ [33]. Lower values were used for IIX, IIA, and I fibers.

## 5. Conclusions

Our mathematical, comprehensive model allows us to gain insight into the kinetics of the Ca^2+^ transients obtained with the fast Ca^2+^ dye Mag-Fluo-4, for the continuum of skeletal muscle fiber types. Sarcoplasmic peak [Ca^2+^] is one order of magnitude higher than reported with slow dyes. The magnitudes of change of the Ca^2+^-bound forms of the Ca^2+^ buffers studied follow the order IIB ≥ IIX > IIA > I, except for mitochondrial peak [Ca^2+^] which showed the pattern I >> IIA >> IIX ≥ IIB. The kinetics for fibers IIA and IIX proved to be intermediate between I and IIB fibers, supporting dynamic data. The results may help better quantitate SOCE fluxes and thermal changes in mammalian fiber types in the future and support the use of fast Ca^2+^ dyes for most experimental approaches in skeletal muscle.

## Figures and Tables

**Figure 1 ijms-22-12378-f001:**
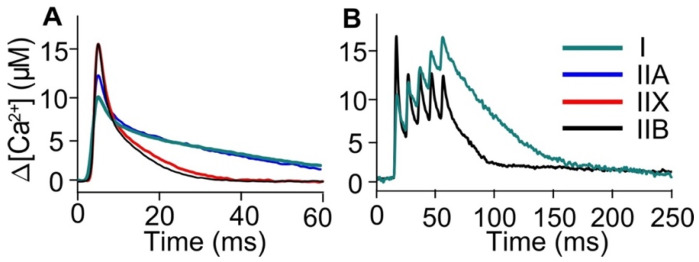
Experimental, calibrated measurements of [Ca^2+^] during Ca^2+^ transients elicited by a single and a train of AP (100 Hz) in mouse fibers at room temperature (21–23 °C). All fibers were classified by myosin heavy chain direct determination in the original paper. Fluorescence recordings generated with Mag-Fluo-4 were used to obtain the ∆[Ca^2+^] in the sarcoplasm produced by a single AP in fiber types I (green), IIA (blue), IIX (red), and IIB (black) (**A**) and a train of 5 AP in fibers I and IIB (**B**).

**Figure 2 ijms-22-12378-f002:**
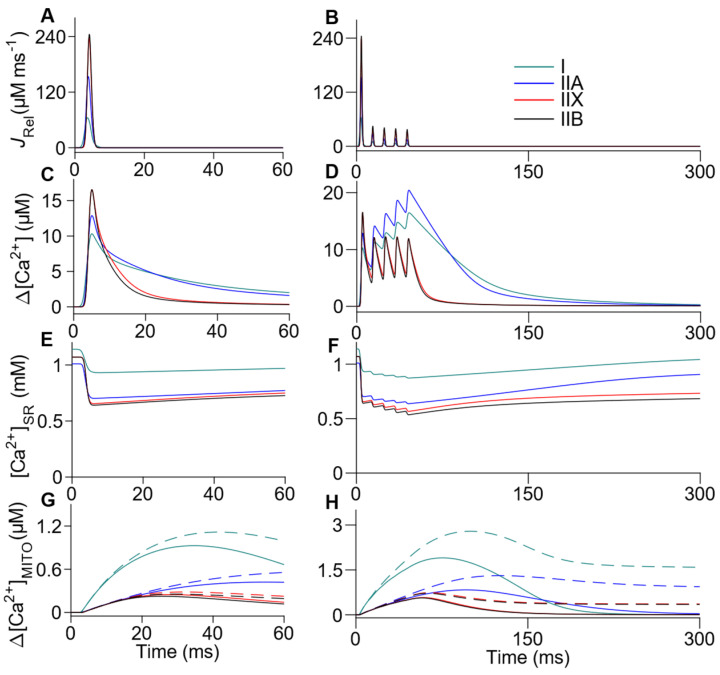
Simulation of release rate of Ca^2+^ and free [Ca^2+^] in the three compartments of the model. The release rate of Ca^2+^ was estimated from the measurements of ∆[Ca^2+^] in fibers I, IIA, IIX, and IIB (**A**). The model was used to reproduce the Δ[Ca^2+^] obtained experimentally in the sarcoplasm (**C**,**D**) and to estimate the Δ[Ca^2+^] in the SR (**E**,**F**), and MITO (**G**,**H**, solid line). The release rate of Ca^2+^ was used in the first AP of the tetanic Ca^2+^ transient simulation and in subsequent AP rescaled to match the experimental Δ[Ca^2+^] peaks (**B**). The tetanic Ca^2+^ transient of IIA and IIX fibers were simulated assuming that they share morphology with I and IIB fibers, respectively. The total [Ca^2+^] in MITO during the Ca^2+^ transients (**G**,**H**, dashed lines), was obtained from the sum of Δ[Ca^2+^]_MITO_ and the Ca^2+^ bound to B. MITO simulations were extended in time for the tetanic transients in the Appendix A, in order to show the return of [Ca^2+^] to the basal levels.

**Figure 3 ijms-22-12378-f003:**
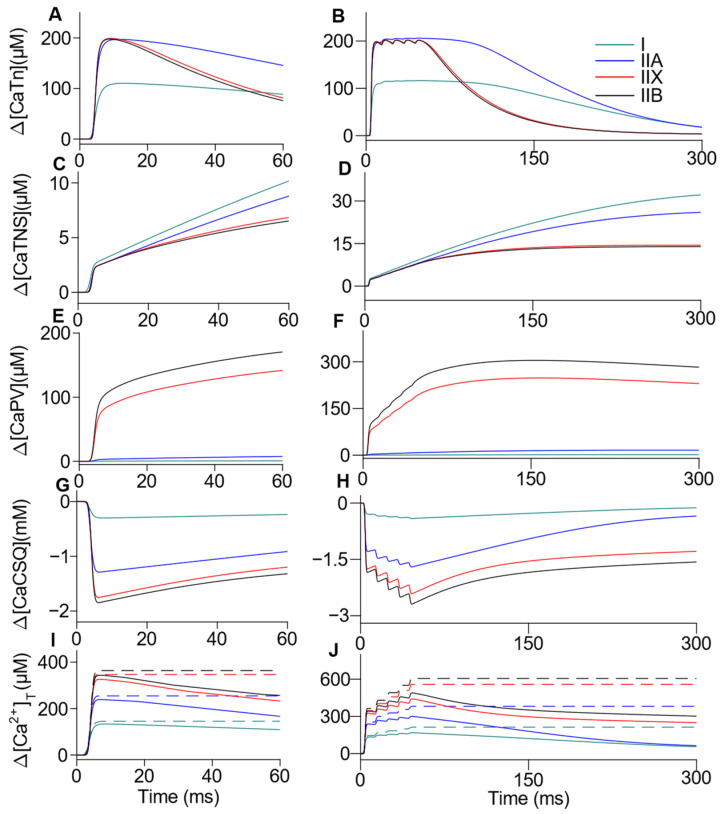
Simulation of single and tetanic Ca^2+^ transients buffering. The Δ[Ca^2+^] coupled to the sarcoplasmic buffers: Tn (**A**,**B**), non-specific sites (TNS) (**C**,**D**), and PV (**E**,**F**); and the SR buffer: calsequestrin (CSQ) (**G**,**H**) is shown for single (left column) and tetanic (right column) transients. The total [Ca^2+^] in the sarcoplasm (solid line), was calculated as the sum of the Δ[Ca^2+^] in both free and bound forms, whereas the total [Ca^2+^] released (dashed line), is the numerical integration of *J*_Rel_ (**I**,**J**).

**Figure 4 ijms-22-12378-f004:**
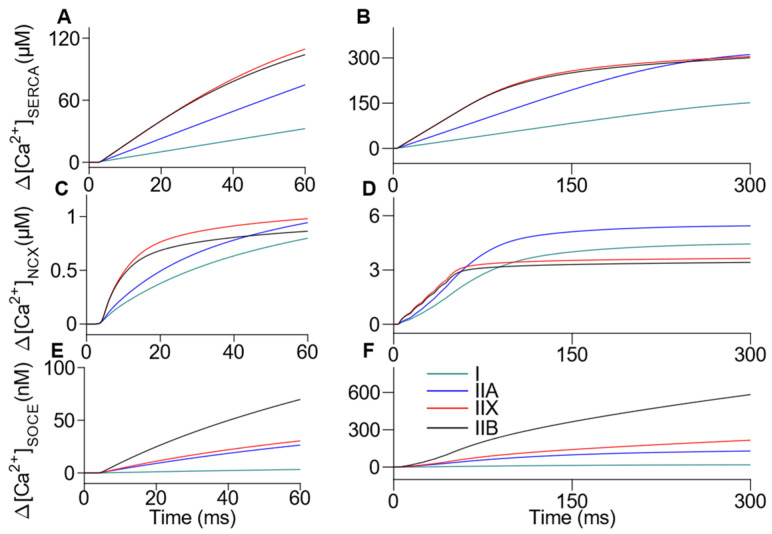
Simulation of kinetics of Ca^2+^ reuptake by the SR, and movement across the sarcolemma during single and tetanic Ca^2+^ transients in the continuum of fiber types. The [Ca^2+^] released is recaptured from the sarcoplasm to the SR by the SERCA (**A**,**B**), extruded from the sarcoplasm to the extracellular space through the NCX (**C**,**D**), and returned from the extracellular space to the sarcoplasm through the SOCE (**E**,**F**) in single (left column) and tetanic (right column) transients.

**Table 1 ijms-22-12378-t001:** List of parameters used for the multi-compartment model simulations of Ca^2+^ transients in skeletal muscle fiber types I, IIA, IIX, and IIB ^a^.

Parameter	Description	Fiber Type		References
		IIA, IIX, and IIB	I	
**Sarcomeric volumes**				
*V* _SR_	Fiber volume occupied by the SR	5.5%	2.9%	[18]
*V* _T-tubule_	Fiber volume occupied by the T-tubule	0.4%	0.2%	[18]
*V* _mito_	Fiber volume occupied by MITO	8.5%	15.4%	[19]
*V* _MS_	Fiber volume occupied by the MS	75%	81%	[20]
*V* _sarc*_	Fiber volume occupied by the sarcoplasm	85.6%	81.5%	(Present study)
**Concentrations**				
[Ca^2+^]_rest,sarc_	Resting free Ca^2+^ in sarcoplasm	106 nM	106 nM	[21]
[Ca^2+^]_rest,SR_	Resting free Ca^2+^ in SR	1.01 mM	1.14 mM	[22]
[Mg^2+^]_rest,sarc_	Resting free Mg^2+^ in sarcoplasm	0.78 mM	0.78 mM	[23]
[Mg^2+^]_T_	Total [Mg^2+^]	3300 µM	3300 µM	[24]
[Na^+^]_sarc_	Sarcoplasmic [Na^+^]	10 mM	10 mM	[25]
[Na^+^]_mito_	Mitochondrial [Na^+^]	5 mM	5 mM	[26]
[Na^+^]_extra_	Extracellular [Na^+^]	140 mM	140 mM	[25]
[Ca^2+^]_extra_	Extracellular [Ca^2+^]	1 mM	1 mM	[25]
[Tn]_T_	Total [Tn]	240 µM	120 µM	[13]
[TNS]_T_	Total [TNS]	240 µM	240 µM	[13]
[PV]_T_	Total [PV]	1900–60 µM	6 µM	[7,9,27]
[ATP]_T_	Total [ATP]	8 mM	5 mM	[28]
[CSQ]_T_	Total [CSQ]	46 mM	23 mM	[27]
[B]_T_	Total [B]	20 µM	20 µM	[14]
**SERCA**				
*V* _SERCA_	Maximum flux rate for SERCA	2.5–1.4 µM ms^−^^1^	0.6 µM ms^−^^1^	(Present study)
*K* _SERCA_	SERCA half-maximum pump [Ca^2+^]	0.44 µM	0.38 µM	[29]
*h* _SERCA_	SERCA Hill coefficient	2.1	2.2	[29]
**MCU**				
*V* _MCU_	Maximum flux rate MCU	18.2 µM s^−^^1^	74.3 µM s^−^^1^	[13,16] and (Present study)
*K* _MCU_	MCU half-maximum pump [Ca^2+^]	1.2 µM	1.97 µM	[30]
*h* _MCU_	MCU Hill coefficient	2	3.5	[30]
**NCE**				
*V* _NCE_	Maximum flux rate NCE	2.25 µM s^−^^1^	9.19 µM s^−^^1^	(Present study)
*K* _NCE,Ca^2+^_	Ca^2+^ binding constant of NCE	1.1 mM	1.1 mM	[14]
*K* _NCE,Na^+^_	Na^+^ binding constant of NCE	8.2 mM	8.2 mM	[31]
ΔΨ_m,mito_	Mitochondrial membrane potential	190 mV	190 mV	[31]
**NCX**				
*V* _NCX_	Maximum flux rate NCX	214.5–120.1 µM s^−^^1^	93.2 µM s^−^^1^	(Present study)
*K* _NCX,Ca^2+^_	Ca^2+^ binding constant of NCX	140 µM	130 µM	[32]
*K* _NCX,Na^+^_	Na^+^ binding constant of NCX	14 mM	11 mM	[32]
ΔΨ_m_	Sarcolemmal membrane potential	80 mV	80 mV	[25]
**SOCE**				
*V* _SOCE_	Maximum flux rate for SOCE	35–19 µM s^−^^1^	15 µM s^−^^1^	[33] and (Present study)
*K* _SOCE_	SOCE half-maximum pump [Ca^2+^]	0.35 mM	0.35 mM	[33]
*h* _SOCE_	SOCE Hill coefficient	4.7	4.7	[33]
**V*_sarc_ was calculated as 100%-*V*_SR_-*V*_T-tubule_-*V*_mito_.				

^a^ The concentrations of free Ca^2+^ in the sarcoplasm and of the binding sites Tn, PV, ATP, and dye are referred to the sarcoplasmic water volume. The free Ca^2+^ in the SR and CSQ are relative to the SR volume. The free Ca^2+^ in the MITO and B are relative to the MITO volume. When required, the vales of the parameters taken from the literature were adjusted to 22 °C. SR: sarcoplasmic reticulum. MITO: mitochondria. MS: myofibril space. Tn: troponin. TNS: troponin non-specific sites. PV: parvalbumin. ATP: adenosine triphosphate. CSQ: calsequestrin. B: total MITO buffers. SERCA: sarcoendoplasmic reticulum Ca^2+^ adenosine triphosphatase. MCU: mitochondrial Ca^2+^ uniporter. NCE: mitochondrial Na^+^/Ca^2+^ exchanger. NCX: Na^+^/Ca^2+^ exchanger. SOCE: store-operated Ca^2+^ entry. *V* indicates maximum capacity of the mechanism. *K* indicates the concentration which induces half-maximal activation of the mechanism.

**Table 2 ijms-22-12378-t002:** Estimated kinetic parameters of the Ca^2+^ release rate function ^a^.

Fiber Type	I	IIA	IIX	IIB
Peak amplitude (µM ms^−1^)	64.8	153.6	238.8	244.5
10–90% Rise time (ms)	1.7	1.2	1.2	1.2
Half-width (ms)	1.9	1.4	1.3	1.3
90–10% Decay time (ms)	2.4	1.8	1.5	1.5
*f*_Rel_ (2nd AP)	0.17	0.18	0.18	0.17
*f*_Rel_ (3rd AP)	0.10	0.11	0.17	0.15
*f*_Rel_ (4th AP)	0.10	0.11	0.16	0.15
*f*_Rel_ (5th AP)	0.10	0.10	0.15	0.14

^a^*f*_Rel_ is the multiplication factor used in the second and subsequent action potentials (APs) to fit the measured peak amplitudes.

**Table 3 ijms-22-12378-t003:** Maximum [Ca^2+^] variations reached during the simulated time intervals of single (60 ms) and tetanic (300 ms) Ca^2+^ transients of I, IIA, IIX, and IIB fiber types ^a^.

Fiber Type	I	IIA	IIX	IIB	I	IIA	IIX	IIB
	Single				Tetanic			
∆[Ca^2+^] (µM)	10.31	12.89	16.55	16.56	16.47	20.44	16.55	16.56
[Ca^2+^]_SR_ (mM)	0.93	0.7	0.65	0.64	0.87	0.64	0.57	0.54
∆[Ca^2+^]_MITO_ (µM)	0.93	0.42	0.25	0.23	1.9	0.83	0.58	0.56
∆[CaB] (µM)	0.33	0.14	0.08	0.07	1.61	0.9	0.36	0.34
∆[CaTn] (µM)	110.23	197.28	199.09	197.75	116.39	206.05	202.16	201.85
∆[CaTNS] (µM)	10.17	8.79	6.84	6.52	32.15	25.99	14.42	13.92
∆[CaPv] (µM)	0.78	7.63	141.78	170.66	1.76	16.27	247.96	304.5
∆[CaATP] (µM)	23.34	46.61	59.75	59.79	37.16	73.65	59.75	59.79
∆[CaDye] (µM)	3.23	4.02	5.14	5.14	5.11	6.31	5.14	5.14
∆[Ca^2+^]_T_ (µM)	133.69	238.93	325.8	343.28	168.29	300.92	441.78	489.53
∆[Ca^2+^]_Rel_ (µM)	145.48	254.58	346.68	363.71	213.43	381.58	558.4	606.1
∆[CaCSQ] (mM)	−0.3	−1.29	−1.75	−1.85	−0.41	−1.7	−2.42	−2.69
∆[Ca^2+^]_SERCA_ (µM)	32.48	74.84	109.39	103.96	151.42	311.23	304.76	300.58
∆[Ca^2+^]_NCX_ (µM)	0.8	0.94	0.98	0.86	4.44	5.44	3.64	3.42
∆[Ca^2+^]_SOCE_ (nM)	3.29	26.39	30.44	69.73	18	129	215.9	582.63

^a^ Abbreviations as in Table 1.

## Data Availability

The data presented in this study are available on request from the corresponding author.

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
