# Peer review of "Comprehensive Simulation of Ca2+ Transients in the Continuum of Mouse Skeletal Muscle Fiber Types"

_ijms, 2021, doi:10.3390/ijms222212378_

Round 1
Reviewer 1 Report
The manuscript analyzes the time varying [Ca2+] simulated by a model with three different compartments (SR, cytosol, and mitochondria) and in four different mammalian fiber types (I, IIA, IIX and IIB). The manuscript goes beyond previous modeling efforts considering four fiber types, which is an interesting novelty. Moreover, it extends previous compartmental models introducing calcium transport outside and inside the cell, introducing somehow a fourth compartment, which is also a critical aspect often neglected in previous models. Moreover, the model is based on experimental data obtained by a reliable and fast dye, which allows for rapid detection of cytosolic calcium transients. The results support the idea of a continuous modification among different fiber types, following the order IIB≥IIX>IIA>I. For these reason, I found this manuscript interesting and worth of publication.
However, some crucial information are missing. These limits, reported below, did not allowed me to fully evaluate the model and must be solved before publication.
Major concern about the model description:
- First, it is needed a more detailed and structural description of the model (eventually as supplementary informations if it is too long for the main text). Having a single compartment for the sarcoplasm, the number of equations are not so much and they can be fully expressed, starting from a mathematical description of what said in the lines 293-295.
- It should be made more clear how many free parameters are used in the model to fit the data, and which data is used to define these parameters. Following Table 1, I see five parameters defined by “present study”. V_serca, V_MCU, V_NCE, V_NCX, V_SOCE. Are those all free parameters? How are they defined? In lines 184-185 the authors states that experimental data are from “present study and [15]”. Please explain in the material and method section which are from the current study.
- Equation (1) is confusing to me. t is time, so T_i should also be a time. Why it is called location? How “tau is associated to the peak width” and how is N determined (what is its value and why)? Why it’s needed both R_i and f_Rel.
- What are the values used for the rate constants k_- and k_+ for all the buffers? How are they defined? Also, I suggest to modify the terms “forward” and “reverse” in line 306 with something more meaningful like binding\unbinding or on rate\off rate. What does it means “different types” in line 307? Must be clarified.
- Please include equations for the SERCA in section 4.2.4. Also explicitly describe and justify the difference mentioned in line 218.
-Line 350: I am not able to retrive the values 0.012 from the references, I would like to have more details about this choice. Also, both the references 11 and 14 include a description of the calcium diffusion inside the sarcoplasm and account for the mitochondria micro domains. The model presented here does not. At least some comments must be included here: how the parameters valid for a diffusional model can be used\adapted to a monocompartimental model of the sarcoplasm?
- How long does it takes for the calcium distribution to come back to the relaxed steady state after AP(s) end? For example in figure 2 H for the calcium bound to the mito buffer, it seems very stable also after 300 ms (by the way some curves should be dashed I think). The same is in figure 3 C or D. I ask the authors to add a longer time analysis as a supplementary figure and add some comments about the kinetics to reach the relaxed steady state.
- As supplementary data, the authors should also include an analysis of the constancy in time of the total calcium amount (not concentration), to show the reliability of the model. Possibly with a description of the distribution of such amount in relaxed situation and at the peak after the AP(s).
Major concern about the calcium concentrations and fluxes:
The authors correctly described the positive aspects of the faster dyes in estimating calcium concentration. However, I strongly suggest to be more cautious in all the statements about the estimation of the peak of calcium concentrations. It is known in literature that this is an open debate. Examples of lower [Ca2+] estimations, apart for the papers mentioned in the manuscript, are, for instance, Allen-Westerblad J Physiol 1994 (Indo-1), or Selin Renaud AJP del 2015 (Fura-2). I suggest to include a deeper and more critical revision of the literature on this aspect, if the authors want to support a strong position in favor of higher calcium concentration. For instance, from studies on skinned fibers exposed to known concentration of calcium, it has been shown that at pCa 6 muscle fibers generate about 50% of the maximum force, and that at less than pCa 5 there is full activation. How can this fit with a 10-15 uM of free calcium (pCa 4.8) after one AP? Similarly, the Tn seems to be almost saturated from figure 3. How the authors explain its relationship with the force exerted? A comment is needed.
- in ref [20] I see that the values used in this paper are indicated as “endogenous Ca2+ content at resting [Ca2+]” in the SR, so 1.01 mM is the total amount if calcium (per liter of fiber) present in the SR and not it’s local free calcium concentration (estimated to be 1.2 mM in Fryer, Owen, Lamb & Stephenson, 1995). Is it an error? Of not, please explain better the choice.
- Please pay more attention to distinguish between local concentration (microm) in organelles and concentration in fiber (micromoles per liter fiber). In the present version of the manuscript this aspect remains somewhat ambiguous
Minor points:
- The importance of SCOE dynamics has deeply been shown by the works of Protasi and colleagues and I suggest that they should be acknowledged in lines 34-37.
- Please, clarify why the occupancy fractions (line 317) must be calculated and not simply obtained from the simulated data. Does it means “imposed”?
-Please add a reference for the definition of the TNS concentration (line 236)
- Some acronyms must be defined at the first appearance to make the reading smoother (J_rel, in line 80, CSQ line 86, TNS line 115 and others…)
- The passage from experimental data and simulated data (lines 72-78) should be clearly stated as well as how the simulation is done, which experimental curves are used to define the parameters and so on. This would make the reading more straightforward.
- line 86 “cero” should be “zero”
- line 84, I would not say that the effect of the CSQ in the model is interesting, but necessary. I would erase the whole sentence.
-Line 120: It is not “flux”, but quantities (integral of the flux in time), I think. Please check.
- line 126: what’s the units for the values “41-121 lower”? Times?
- line 138: “Maximum [Ca2+] and minimum in the SR” is not clear, please rephrase.
- Line 162: I do not see the dichotomy approach as a limit of the previous models. They are pretty stable in describing two specific fiber types. It is not required to “go beyond” in my opinion. I suggest to rephrase the sentence stating that the current model analyze the four fiber types, being this a novelty.
-Line 255: I ask for a more detailed explanation of the “half-with differences in the recordings that fed the simulations”. Apparently it is an important aspect to be defined in the paper. How the recordings fed the simulations?
- Line 284 and 285: I would give at least some insight about the method used.
-Line 322: Tn concentration is 120 uM not mM, please correct
Author Response
Medellín, Oct 1st, 2021
Ms. Milica Lajic
Assistant Editor
International Journal of Molecular Sciences
MDPI
Dear Editor
We present a detailed list of modifications done to the manuscript “Comprehensive simulation of Ca²⁺ transients in the continuum of mouse skeletal muscle fiber types”, with ID: ijms-1391248, according to the reviewer’s suggestions. Our answers are indented. Changes in the main manuscript are marked up using the “Track Changes” function.
Reviewer 1
The manuscript analyzes the time varying [Ca2+] simulated by a model with three different compartments (SR, cytosol, and mitochondria) and in four different mammalian fiber types (I, IIA, IIX and IIB). The manuscript goes beyond previous modeling efforts considering four fiber types, which is an interesting novelty. (…) For these reason, I found this manuscript interesting and worth of publication.
However, some crucial information are missing. These limits, reported below, did not allowed me to fully evaluate the model and must be solved before publication.
Major concern about the model description:
-First, it is needed a more detailed and structural description of the model (eventually as supplementary informations if it is too long for the main text). Having a single compartment for the sarcoplasm, the number of equations are not so much and they can be fully expressed, starting from a mathematical description of what said in the lines 293-295.
A description of the model equations system was included in Methods, section 4.2. We feel it fits well the main text. We respectfully refer the reviewer to that section in order not to copy all that information and equations here.
-It should be made more clear how many free parameters are used in the model to fit the data, and which data is used to define these parameters.
We added the following sentence in Methods, section 4.2: “The parameters that describe the JRel, and the values of maximum capacity of SERCA, NCX, NCE and SOCE (VSERCA, VNCX, VNCE and VSOCE), were fitted to the measured Ca2+ transients. All other values of the model were taken from the literature, and the specific references are given in Table 1.”
Following Table 1, I see five parameters defined by “present study”. V_serca, V_MCU, V_NCE, V_NCX, V_SOCE. Are those all free parameters?
VSERCA, VNCX, VNCE and VSOCE were free parameters used in the model to fit the data. We made this clearer in Section 4.2, as mentioned above. VMCU was taken from references 13 and 16 and adjusted to a temperature of 22°C. This adjustment was acknowledged in the legend of Table 1.
How are they defined?
In enzymatic kinetics, V refers to maximum capacity, so we used the same symbol to indicate the maximum capacity of the different mechanisms. In section 4.2 we indicated that VSERCA, VNCX VNCE and VSOCE refer to the maximum capacity of SERCA, NCX, NCE and SOCE, respectively and refer the reader to Table 1. These mechanisms and abbreviations were already described in the introduction, and are further developed throughout the Methods section. VSERCA was fitted to the decay phase of the Ca2+ transients (now mentioned in section 4.2.4). VNCE was fitted to the decay phase time of the mitochondrial [Ca²⁺] (now mentioned in section 4.2.5). The values of VNCX were obtained considering VSERCA and other experimental information, which is now mentioned in section 4.2.6. To define VSOCE values we considered that the fastest fibers achieved up to 35 μM s-1, as seen in Figure 3C of Koenig et al. 2019 (doi:10.1016/j.bbamcr.2019.02.014), and that the Ca2+ extruded by NCX seems to be recycled by SOCE. This is now mentioned in section 4.2.7. The values of VMCU were taken from the references 12 and 15, however, those references considered a temperature of 16°C and we adjusted the values to a temperature of 22°C.
In lines 184-185 the authors states that experimental data are from “present study and [15]”. Please explain in the material and method section which are from the current study.
We apologize for leading to a confusion. With “present study” we specifically refer to the calibration procedure performed in the present manuscript. Such calibration was according to our recent paper (doi:10.1016/j.bbagen.2021.129939). To make this clearer, we completed the lines 284 and onwards, of the Methods section: “The conversion of fluorescence intensity of single and tetanic experimental recordings to [Ca²⁺] was performed according to the calibration method and the value parameters presented in [16]”. In line 184 we made clear that we refer to the “Calibration performed in the present study and [16]”
- Equation (1) is confusing to me. t is time, so T_i should also be a time. Why it is called location?
We rephrased the sentence in line 298 and onwards to avoid confusion.
How “tau is associated to the peak width” and how is N determined (what is its value and why)? why it’s needed both R_i and f_Rel.
We estimated the release rate of single and tetanic transients, i.e., recordings with one or more peaks, and N determines the number of peaks used to fit the release rate produced by one or more APs. That peak or those peaks have an amplitude, which is considered by Ri. For tetanic transients, since the amplitude of the second and subsequent peaks is reduced compared to the peak produced by the first AP, a multiplication factor fRel is used to rescale those amplitudes. The description of these parameters is presented in lines 298-302.
-What are the values used for the rate constants k_- and k_+ for all the buffers? How are they defined?
The rate constants values are now included in the Supplemental Table 1.
Also, I suggest to modify the terms “forward” and “reverse” in line 306 with something more meaningful like binding\unbinding or on rate\off rate.
We followed the suggestion of the referee and the terms were changed to binding and unbinding.
What does it means “different types” in line 307? Must be clarified.
To avoid confusion, the sentence was rephrased as: “The interactions of Ca²⁺ and Mg²⁺ with the binding sites were described as reversible reactions”.
-Please include equations for the SERCA in section 4.2.4.
The equation for the SERCA was included in section 4.2.4, as requested.
Also explicitly describe and justify the difference mentioned in line 218.
The differences arise from the temperature corrections. This changes the kinetics of the reactions, which is expected to also change how other factors affect the Ca2+-Tn reaction. That is why we mentioned the inclusion of a larger SERCA capacity, MITO and SOCE as other regulators of this reaction, which may reduce the amount of [CaTn] that we found. We complemented the sentence in line 218 to mention the temperature corrections.
-Line 350: I am not able to retrive the values 0.012 from the references, I would like to have more details about this choice. Also, both the references 11 and 14 include a description of the calcium diffusion inside the sarcoplasm and account for the mitochondria micro domains. The model presented here does not. At least some comments must be included here: how the parameters valid for a diffusional model can be used\adapted to a monocompartimental model of the sarcoplasm?
The value 0.012 was taken from the reference 14, in the subsection Possible Influence of Ca2+ Uptake by Mitochondria (page 298). The value 0.049 was taken from the reference 11, in the subsection Possible role of Ca2+ uptake by mitochondria (page 592). Although references 11 and 14 consider the Ca2+ diffusion, those models did not consider the mitochondrial microdomains. The analysis presented in both references for the Ca2+ uptake by the mitochondria did not consider diffusion and no adaptations were required to apply them in our work.
-How long does it takes for the calcium distribution to come back to the relaxed steady state after AP(s) end? For example in figure 2 H for the calcium bound to the mito buffer, it seems very stable also after 300 ms (by the way some curves should be dashed I think). The same is in figure 3 C or D. I ask the authors to add a longer time analysis as a supplementary figure and add some comments about the kinetics to reach the relaxed steady state.
We thank the referee for the suggestion. Supplementary Figure 1 with a longer time interval was included for some reactions, which did not reach their relaxed steady state during the time considered in the manuscript.
-As supplementary data, the authors should also include an analysis of the constancy in time of the total calcium amount (not concentration), to show the reliability of the model. Possibly with a description of the distribution of such amount in relaxed situation and at the peak after the AP(s).
We have followed the referee’s suggestion and included the following analysis at the end of the Results section: “The total amount of Ca²⁺ obtained at rest considering all compartments of the model, including the extracellular space, remained constant during the simulated activation interval. We obtained that the variations in the total amount of Ca²⁺ were lower than 10-6 µM. This result evidences that truncation errors were negligible during the simulations”.
Major concern about the calcium concentrations and fluxes:
The authors correctly described the positive aspects of the faster dyes in estimating calcium concentration. However, I strongly suggest to be more cautious in all the statements about the estimation of the peak of calcium concentrations. It is known in literature that this is an open debate. Examples of lower [Ca2+] estimations, apart for the papers mentioned in the manuscript, are, for instance, Allen-Westerblad J Physiol 1994 (Indo-1), or Selin Renaud AJP del 2015 (Fura-2). I suggest to include a deeper and more critical revision of the literature on this aspect, if the authors want to support a strong position in favor of higher calcium concentration. For instance, from studies on skinned fibers exposed to known concentration of calcium, it has been shown that at pCa 6 muscle fibers generate about 50% of the maximum force, and that at less than pCa 5 there is full activation. How can this fit with a 10-15 uM of free calcium (pCa 4.8) after one AP? Similarly, the Tn seems to be almost saturated from figure 3. How the authors explain its relationship with the force exerted? A comment is needed.
Thank you for rising this important issue. We have addressed it before, for instance in the review Calderón et al. The excitation-contraction coupling mechanism in skeletal muscle. Biophys Rev. 2014 Mar;6(1):133-160. doi: 10.1007/s12551-013-0135-x, and in our recent paper Milán et al. Calibration of mammalian skeletal muscle Ca2+ transients recorded with the fast Ca2+ dye Mag-Fluo-4. Biochim Biophys Acta Gen Subj. 2021 Sep;1865(9):129939. doi: 10.1016/j.bbagen.2021.129939.
We are aware of the work of several important groups in the area, which have used or still use Fura-2 and Indo-1; for instance, those in France (group of Vincent Jacquemond), Sweden (Håkan Westerblad and Joseph Bruton), Italy (Carlo Reggiani, Feliciano Protasi), Switzerland or USA. Their works have already been quoted.
We sincerely accept the call of the reviewer to be more cautious about this issue in the present manuscript, whose objective is not to evaluate the performance of fast vs. slow dyes in skeletal muscle. Also, we do not want to explicitly write against any specific group or author. Thus, we modified the sentence in line 192 to: “The above numbers are one order of magnitude higher than those reported with slow dyes [12,31,41,42]”. Also, the subtitle was changed to: “3.2 [Ca2+] kinetics with a fast Ca2+ dye”.
Finally, we included two implications of the higher [Ca2+] to ECC issues which is the focus of our work, for instance in the driving force for SOCE or the thermal changes related to Ca2+ fluxes. The issue of the [Ca2+] and force goes beyond the peak of Ca2+, since the studies of Edman suggested that the duration of the Ca2+ transient is also relevant. Thus, we consider that the issue of the reliability of the measurement with fast vs slow dyes, and their relation with force, deserves a deep scrutiny in a future review, for instance.
- in ref [20] I see that the values used in this paper are indicated as “endogenous Ca2+ content at resting [Ca2+]” in the SR, so 1.01 mM is the total amount if calcium (per liter of fiber) present in the SR and not it’s local free calcium concentration (estimated to be 1.2 mM in Fryer, Owen, Lamb & Stephenson, 1995). Is it an error? Of not, please explain better the choice.
The value of 1.01 mM was estimated as the Ca2+ inside the SR in reference 20. However, this is the free Ca2+, since the total Ca2+ includes the amount bound to CSQ. Not all Ca2+ is free to be released, this is why the models include the free Ca2+ as the most important parameter regarding Ca2+ amounts for modeling single transients. The amount of Ca2+ free and bound to CSQ was taken into account when modeling tetanic transients.
-Please pay more attention to distinguish between local concentration (microm) in organelles and concentration in fiber (micromoles per liter fiber). In the present version of the manuscript this aspect remains somewhat ambiguous.
Thank you for the observation. The following sentence was included in the caption of Table 1 to address this concern: “The concentrations of free Ca²⁺ in the sarcoplasm and of the binding sites Tn, PV, ATP and Dye are referred to the sarcoplasmic water volume. The free Ca²⁺ in the SR and CSQ are relative to the SR volume. The free Ca²⁺ in the MITO and B are relative to the MITO volume.”
Minor points:
-The importance of SCOE dynamics has deeply been shown by the works of Protasi and colleagues and I suggest that they should be acknowledged in lines 34-37.
Thank you for the suggestion. We now quoted these two nice papers by Feliciano Protasi and colleagues: Calcium entry units (CEUs): perspectives in skeletal muscle function and disease - PubMed (nih.gov), and Orai1-dependent calcium entry promotes skeletal muscle growth and limits fatigue - PubMed (nih.gov)
- Please, clarify why the occupancy fractions (line 317) must be calculated and not simply obtained from the simulated data. Does it means “imposed”?
The reviewer is right and helped us to be more concise and precise. The lines 317 to 320 were simplified as: “The occupancy fraction of the binding sited at equilibrium were obtained from the simulated data”.
-Please add a reference for the definition of the TNS concentration (line 236)
Thanks for the suggestion. The following reference was included in the discussion: “The TNS have peaked 31 µM during a train of APs in slow fiber types after 200 ms [12], and in the present work, such as with Tn, have peaked less, being 22 µM. Specifically in type IIX and IIB fibers, which have a higher quantity of PV, their influence is reduced, since the binding and unbinding rates are similar to those of PV”.
- Some acronyms must be defined at the first appearance to make the reading smoother (J_rel, in line 80, CSQ line 86, TNS line 115 and others…)
We apologize for this minor problem, which is due to the journal’s template with methods at the end. The acronyms are now defined at their first appearance. Also, the caption of the Table 1 was completed with the explanation of all acronyms used in it.
-The passage from experimental data and simulated data (lines 72-78) should be clearly stated as well as how the simulation is done, which experimental curves are used to define the parameters and so on. This would make the reading more straightforward.
Figure 1 corresponds to the experimental recordings which fed the simulations presented in Figures 2 to 4. To make this clearer, we changed the beginning of the Results section, lines 68-82, to:
“Results
Experimentally recorded, raw, single and tetanic Mag-Fluo-4 Ca2+ transients were calibrated in order to obtain the ∆[Ca²⁺] in the sarcoplasm (Fig 1). The [Ca²⁺] peaks for the continuum of fiber types were: IIB and IIX: 16.58 µM, IIA: 12.77 µM and I: 10.13 µM (Fig. 1A). For the tetanic Ca²⁺ transients, subsequent peaks were also calculated (I: 11.24, 12.95, 14.81 and 16.47 µM; IIB: 12.13, 12.29, 12.24 and 11.96 µM) (Fig. 1B).
(Please see Figure 1 in the main manuscript)
These calibrated signals fed all next estimations and simulations, as described in Methods section and Table 1. First, we mathematically estimated the release rate of Ca²⁺ (JRel) for both single and tetanic transients (Fig. 2A-B; Table 2) and then simulated the Ca2+ kinetics in the sarcoplasm (Fig. 2C-D), the SR (Fig. 2E-F) and MITO (Fig. 2G-H). Fibers type II peaked higher than fibers type I (~137% higher for IIA and 269-277% higher for IIX and IIB). The JRel estimated in tetanic Ca²⁺ transients shows that the last peak’s amplitude is reduced over 10 times for type I fibers and up to 7 times for IIB, IIX and IIA (Fig. 2B; Table 2). The simulated sarcoplasmic ∆[Ca²⁺] closely resembled the experimental recording described above. The [Ca²⁺]SR rapidly decreased and slowly recovered as expected. Although qualitatively similar for all fiber types, quantitative differences arose mainly between the fibers type I and all fibers type II, in agreement with the fact that the Ca2+ released by the fibers type I was the lowest.”
- line 86 “cero” should be “zero”
Following the next suggestion, the sentence that include this word was deleted.
- line 84, I would not say that the effect of the CSQ in the model is interesting, but necessary. I would erase the whole sentence.
Taking into account the suggestion of the referee, this sentence was removed.
-Line 120: It is not “flux”, but quantities (integral of the flux in time), I think. Please check.
We rephrased the sentence to: “Simulation of kinetics of Ca2+ reuptake by the SR, and movement across the sarcolemma during single and tetanic Ca²⁺ transients in the continuum of fiber types.”
- line 126: what’s the units for the values “41-121 lower”? Times?
Thank you for the observation. “Times” was included accordingly.
- line 138: “Maximum [Ca2+] and minimum in the SR” is not clear, please rephrase.
We rephrased the sentence to avoid confusion. The title of Table 3 now reads as: “Maximum [Ca2+] variations reached during the simulated time intervals of single (60 ms) and tetanic (300 ms) Ca2+ transients of I, IIA, IIX and IIB fiber types.”
- Line 162: I do not see the dichotomy approach as a limit of the previous models. They are pretty stable in describing two specific fiber types. It is not required to “go beyond” in my opinion. I suggest to rephrase the sentence stating that the current model analyze the four fiber types, being this a novelty.
We rephrased the subtitle to: “A model which includes four fiber types”, and the sentence in line 162 to: “By taking as experimental source Ca2+ transients obtained in molecularly typed fibers covering the whole spectra from I to IIB, our model adds novel information on the ECC-fiber types relationship.”
-Line 255: I ask for a more detailed explanation of the “half-with differences in the recordings that fed the simulations”. Apparently it is an important aspect to be defined in the paper. How the recordings fed the simulations?
As suggested by previous reviewer´s comments, we completed some sections mentioning that, for instance, VSERCA was based on the fitting of the decay phase of the Ca2+ transients. Recordings with a slow decay phase are wider at their half amplitude. This width at half amplitude is called half-width in several of our previous papers. If we compare our recordings with those of the group of Steve Baylor (refs 11 and 14 quoted in lines 251 and 253), we can find small differences in the width at half amplitude. Those small differences may explain small differences found in SERCA function between both groups (our and Baylor´s).
- Line 284 and 285: I would give at least some insight about the method used.
We followed the method we recently published in Milán et al. Calibration of mammalian skeletal muscle Ca2+ transients recorded with the fast Ca2+ dye Mag-Fluo-4. Biochim Biophys Acta Gen Subj. 2021 Sep;1865(9):129939. doi: 10.1016/j.bbagen.2021.129939. Thus, to address the suggestion of the reviewer we included a brief description of the method and some relevant values. We kindly refer the reviewer to the Methods section 4.1, in order not to copy all the information and equations here.
-Line 322: Tn concentration is 120 uM not mM, please correct
We apologize for the mistake. The correction was done.
Best regards,
Marco A. Giraldo
Oscar A. Rincón
Juan C. Calderón

Reviewer 2 Report
This ms looks at the movements of Ca during EC coupling across the range of fibre types in skeletal muscle. The novelty lies in the modelling of the 4 fibre types (1 slow and 3 fast fibres) and also the inclusion of Ca movements across the mitochondrial membrane and t-system membrane.
- There is not much data to base the modelling of Ca2+ movements across the mitochondrial membranes. Most Ca measures are uncalibrated, low dynamic range (genetically encoded probes) or performed at low temporal resolution. The Ca buffering in the mitochondria is also not known. Ref 12 cites a cardiac paper of Bers, who also hazard a guess to the B of the mito. Please acknowledge in the text this uncertainty, and perhaps guess how much B in mito may vary. It seems likely that free Ca2+ in mito is in the nM range, so we may expect a large number of B site inside mito to temper changes in free Ca2+ during EC coupling, where the cyto Ca2+ changes drastically.
- The use of the decline in SR Ca2+ during EC coupling to generate a model for how much Ca/rate of Ca entering the cytoplasm may be an underestimate with this approach. STIM1 is a transmembrane protein at the SR terminal cisternae (to activate SOCE), which means that it detect near-membrane Ca, or SR terminal cisternae Ca2+ levels. A new study by Reddish et al 2021 iScience, show that there are gradients of Ca2+ inside the SR during EC coupling, where terminal cisternae/near-membrane Ca decreases much more rapidly than the bulk SR, which is the current variable in the model. Perhaps consider both aspects of Ca inside the SR and use as a discussion point. (as opposed to dodging this as stated in 3.4).
Minor
p.2. line 54: reword this sentence.
p.2. line 86 "cero" should be "zero"?
p. 3. line 110-112. reword sentence.
p. 8. line 206, attended can be 'studied'
p. 8. line 207. surpass can be 'address'.
p.8. line 237 onward, the values of free Ca2+ in mito during EC coupling in Ref 12 are poor calibrated and low dynamic range, and I have low confidence in these values. I would qualify the comments here with these caveats.
p.8. line 241. "This delay..." reword this sentence and remove 'retard'.
p.9 line 261. The Ca extruded by the t-system seems to be immediately recycled, and has been called a 'counter-flux' (Koenig et al 2018). Please reword.
Author Response
Medellín, Oct 1st, 2021
Ms. Milica Lajic
Assistant Editor
International Journal of Molecular Sciences
MDPI
Dear Editor
We present a detailed list of modifications done to the manuscript “Comprehensive simulation of Ca²⁺ transients in the continuum of mouse skeletal muscle fiber types”, with ID: ijms-1391248, according to the reviewer’s suggestions. Our answers are indented. Changes in the main manuscript are marked up using the “Track Changes” function.
Reviewer 2
This ms looks at the movements of Ca during EC coupling across the range of fibre types in skeletal muscle. The novelty lies in the modelling of the 4 fibre types (1 slow and 3 fast fibres) and also the inclusion of Ca movements across the mitochondrial membrane and t-system membrane.
- There is not much data to base the modelling of Ca2+ movements across the mitochondrial membranes. Most Ca measures are uncalibrated, low dynamic range (genetically encoded probes) or performed at low temporal resolution. The Ca buffering in the mitochondria is also not known. Ref 12 cites a cardiac paper of Bers, who also hazard a guess to the B of the mito. Please acknowledge in the text this uncertainty, and perhaps guess how much B in mito may vary. It seems likely that free Ca2+ in mito is in the nM range, so we may expect a large number of B site inside mito to temper changes in free Ca2+ during EC coupling, where the cyto Ca2+ changes drastically.
Thank you for rising this important point. We completed the discussion, section 3.3, considering this uncertainty: “Previous works have estimated the total [B]. From the simulated analysis performed in Wüst et. al (2017) of the total [B] in cardiac muscle fibers a value of 2 µM was found to be optimal [55]. In the work of Marcucci et. al (2018) for skeletal muscle fibers a larger buffer concentration of 20 µM was also tested. In the present study a value of 20 µM was used. A sizeable total [B] inside MITO is expected in order to deal with the larger and faster amounts of Ca2+ released in skeletal compared to cardiac muscle, and to explain the slowing of decay of the Ca2+ transients when the MITO uptake is blunted. However, well calibrated experimental measurements of MITO Ca2+ buffering are still required to reach an accurate estimate of the total [B] in skeletal muscle fibers.”
- The use of the decline in SR Ca2+ during EC coupling to generate a model for how much Ca/rate of Ca entering the cytoplasm may be an underestimate with this approach. STIM1 is a transmembrane protein at the SR terminal cisternae (to activate SOCE), which means that it detect near-membrane Ca, or SR terminal cisternae Ca2+ levels. A new study by Reddish et al 2021 iScience, show that there are gradients of Ca2+ inside the SR during EC coupling, where terminal cisternae/near-membrane Ca decreases much more rapidly than the bulk SR, which is the current variable in the model. Perhaps consider both aspects of Ca inside the SR and use as a discussion point. (as opposed to dodging this as stated in 3.4).
Thanks for the suggestion. We included the following sentence in subsection 3.4 in order to address this comment: "Single-compartment simulations are expected to have errors associated with an inability to estimate local gradients in [Ca2+] and in the [Ca2+] bound to the binding sites. In the present study the SOCE flux is associated with changes in [Ca2+] in the SR compartment. However, the Ca2+ gradients in the SR have been measured during ECC and the Ca2+ levels decrease more rapidly in the terminal cisternae than in other regions of the SR [57]. Therefore, a model that consider the gradients in the SR would allow a more accurate estimate of the amount of calcium entered by the SOCE.”
Minor
p.2. line 54: reword this sentence.
This line was rewritten as: “However, that model was performed based on Ca²⁺ transients' measurements with Fura-2"
p.2. line 86 "cero" should be "zero"?
Following the suggestion of the reviewer 1, the sentence that include this word was deleted.
- 3. line 110-112. reword sentence.
The sentence now reads like: “The ∆[CaDye] explains only 2.7% of the intracellular dye in the fiber type IIA and 2.2% in fiber types I, IIX and IIB, thus ensuring dye unsaturation.”
- 8. line 206, attended can be 'studied'
Thank you for the suggestion. The change was made.
- 8. line 207. surpass can be 'address'.
Thank you for the suggestion. The change was made.
p.8. line 237 onward, the values of free Ca2+ in mito during EC coupling in Ref 12 are poor calibrated and low dynamic range, and I have low confidence in these values. I would qualify the comments here with these caveats.
We do agree with the reviewer in that the measurements across mitochondrial membranes or inside mitochondria are affected by several shortcomings.
In the introduction we called the attention to the fact that results in ref 12 were obtained with Fura-2, which may have limited the quality of their results. However, we reached MITO values not much different from those reported in that paper. And what is more interesting is that the values of ref 12 and ours coincide (for fast fibers) with a third paper by a third research group, quoted as ref 45. Thus, we believe that specifically criticing ref 12 would be contradictory.
Therefore, in order to acknowledge these issues in general, we included this sentence in section 3.3 “However, well calibrated, high resolution, experimental measurements of MITO Ca2+ buffering and kinetics are still required to reach an accurate estimate of the total [B] and [Ca2+] inside MITO in skeletal muscle fibers.”
p.8. line 241. "This delay..." reword this sentence and remove 'retard'.
Thanks for the suggestion, the sentence was rephrased as: “This delay likely reflects Ca2+ diffusion into MITO, as experimentally measured in mouse using genetically encoded sensors [54].”
p.9 line 261. The Ca extruded by the t-system seems to be immediately recycled, and has been called a 'counter-flux' (Koenig et al 2018). Please reword.
We rephrased the sentence in order to take the reviewer suggestion into account.
We greatly appreciate the thoughtful comments and suggestions of the reviewers, as they helped us improving our manuscript.
Best regards,
Marco A. Giraldo. PhD Biophysics Group |
Oscar A. Rincón Biophysics Group |
Juan C. Calderón. PhD PHYSIS Group University of Antioquia Medellin, Colombia |

Round 2
Reviewer 1 Report
The authors addressed properly all the observations.